chemical biology

cyclopentanocucurbituril, amino acids, self-assembly, inclusions

**Author for correspondence:**
Peihua Ma
e-mail: phma@gzu.edu.cn

†These authors contributed equally to this study.

This article has been edited by the Royal Society of Chemistry, including the commissioning, peer review process and editorial aspects up to the point of acceptance.

# The binding behaviours between cyclopentanocucurbit[6]uril and three amino acids

Siyuan Cheng[1,†], Weiwei Zhao[1,†], Xinan Yang[1],
Ye Meng[1], Liantong Wei[2], Zhu Tao[1] and Peihua Ma[1]

[1]Key Laboratory of Macrocyclic and Supramolecular Chemistry of Guizhou Province, Guizhou University, Guiyang 550025, People's Republic of China
[2]Guiyang Bewg Water Co., Ltd., Guiyang 550001, People's Republic of China

PM, 0000-0002-4965-0632

Binding behaviours between cyclopentanocucurbit[6]uril ($CyP_6Q[6]$) and three amino acids have been investigated by means of X-ray crystallography, proton nuclear magnetic resonance spectroscopy and isothermal titration calorimetry. The results showed that $CyP_6Q[6]$ forms a 1:2 inclusion complex with glycine, but 1:1 complexes with both leucine and lysine. Whereas the carboxyl group of glycine can enter the interior of the cavity of $CyP_6Q[6]$, only the alkyl chains of leucine and lysine can enter this cavity. Interestingly, leucine can adopt two different self-assembly modes upon its interaction with cucurbituril, depending on the external conditions, whereas glycine and lysine do not exhibit such behaviour.

## 1. Introduction

Cucurbit[$n$]urils ($n = 5$–8, 10, 14) are macrocyclic compounds with an inner hydrophobic cavity and two portals rimmed by polar carbonyl oxygen atoms, formed by multiple glycoluril monomers doubly bridged by methylene units [1–3]. As fourth-generation supramolecular hosts following crown ethers, cyclodextrins and calix[$n$]arenes, their high-affinity hydrophobic cavities have propelled their host–guest chemistry into the mainstream, making it a 'hot spot' in cucurbituril chemistry. Because the hydrophobic cavities of cucurbiturils readily form stable inclusion complexes or rotaxane analogues, molecular capsules and other supramolecular structures incorporating various organic small molecules [4–7], especially in aqueous systems, cucurbituril host–guest chemistry has played a significant role in the fields of drug delivery, chemical sensors and cucurbituril supramolecular self-assembly materials [8–10]. Most cucurbiturils show poor solubility in water,

**Scheme 1.** CyP6Q[6] and the amino acids used in this study.

R. Soc. Open Sci. **8**: 202120

except, to some extent, cucurbit[7]uril, which has restricted their applications. The appearance of modified cucurbiturils, especially functionalized cucurbiturils with more water-soluble alkyl substituents and derivatives, has attracted ever more attention. With the emergence of modified cucurbiturils with excellent oil and water solubilities, the host–guest chemistry of cucurbiturils will gradually extend to other organic solvent systems, such as methanol, ethanol solution and dimethyl sulfoxide, and has played a vital role in functional application research [11–13]. Cyclopentylcucurbit[6]uril (abbreviated as CyP$_6$Q[6], scheme 1) is a good oil- and water-soluble derivative, which has broad application prospects. However, as yet, there have been few studies on the synthesis and properties of cyclopentyl cucurbituril, and the research is still in its infancy [14–16].

Amino acids, the key constituents of protein/peptide bonds and important components of living systems, have always attracted much attention [17], and they have been widely studied in the host–guest chemistry of cucurbiturils. Because cucurbit[n]urils have both electronegative carbonyl oxygen-fringed portals and a hydrophobic cavity [18,19], their ion-dipole and hydrophobic effects make them well-suited to form host–guest complexes with amino acids. Buschmann *et al.* and Zhang *et al.* found that cucurbit[6]uril readily binds to some amino acids with hydrophobic side chains [20,21]. Kovalenko *et al.* and Lee *et al.* studied the binding behaviour between cucurbit[7]uril and amino acids with different side chains in the gas and liquid phases, respectively, and observed different binding affinities under different conditions [22,23]. Both Bush *et al.* and Nau *et al.* found that the larger cavity of cucurbit[8]uril can selectively bind a small organic molecule and an amino acid simultaneously [24,25]. Our research group reported the host–guest binding behaviour of twisted cucurbit[14]uril and inverted cucurbit[7]uril with amino acids [26,27], and, to our knowledge, for the first time reported supramolecular complexes of cucurbituril and enantiomeric amino acids [28].

In recent years, studies on the binding behaviour between macrocyclic compounds, such as cyclodextrin and crown ethers, and amino acid molecules have been extremely widespread [29–31]. However, the research on the host–guest chemistry of cyclopentyl cucurbiturils is still immature, so we are interested in the study on the host–guest properties of CyP6Q[6] and amino acids. For the present study, we took CyP$_6$Q[6] as the host, and selected three different amino acids, namely glycine (Gly), L-lysine (L-Lys) and L-leucine (L-Leu), as guests. We examined the binding behaviour between these components in the solid and liquid phases (scheme 1).

## 2. Experimental

### 2.1. Material and methods

All raw materials used in this study were purchased from Aladdin Industrial Corporation (AR, Shanghai, China). CyP$_6$Q[6] was prepared according to a literature procedure [14].

### 2.2. Preparation of complexes **1–4**

#### 2.2.1. Complex **1**

Preparation of CyP$_6$Q[6]@2Gly@[CdCl$_4$]$^{2-}$ crystals. CyP$_6$Q[6] (10 mg, 8.1 µmol), Gly (5.76 mg, 72.9 µmol) and CdCl$_2 \cdot$2H$_2$O (7.2 mg, 32.4 µmol) were added to 3 M HCl (3 Ml), and the solution was boiled for

about 1 min. It was then filtered, and the filtrate was left to stand at room temperature. After several days, single crystals of CyP$_6$Q[6]@2Gly@[CdCl$_4$]$^{2-}$ suitable for X-ray diffraction analysis were obtained in 38% yield.

### 2.2.2. Complex **2**

Preparation of CyP$_6$Q[6]@L-Lys@[CdCl$_4$]$^{2-}$ crystals. CyP$_6$Q[6] (10 mg, 8.1 µmol), L-Lys (10.8 mg, 72.9 µmol) and CdCl$_2$ · 2H$_2$O (7.2 mg, 32.4 µmol) were added to 3 M HCl (3 ml), and the solution was boiled for about 1 min. It was then filtered, and the filtrate was left to stand at room temperature. After several days, single crystals of CyP$_6$Q[6]@L-Lys@[CdCl$_4$]$^{2-}$ suitable for X-ray diffraction analysis were obtained in 42% yield.

In the preparation of inclusion compound L-Leu@CyP$_6$Q[6], after many attempts to use cadmium salts and failure to cultivate the corresponding crystals, we used zinc salts of the same group element and successfully cultivated crystals of complexes **3** and **4**.

### 2.2.3. Complex **3**

Preparation of CyP$_6$Q[6]@L-Leu@[ZnCl$_3$ · H$_2$O]$^-$ crystals. CyP$_6$Q[6] (10 mg, 8.1 µmol), L-Leu (9.6 mg, 72.9 µmol), and ZnCl$_2$ · 2H$_2$O (5.6 mg, 32.4 µmol) were added to 2.5 M HCl (3 ml), and the solution was heated for about 1 min. It was then filtered, and the filtrate was left to stand at room temperature. After several days, single crystals of CyP$_6$Q[6]@L-Leu@[ZnCl$_3$ · H$_2$O]$^-$ suitable for X-ray diffraction analysis were obtained in 32% yield.

### 2.2.4. Complex **4**

Preparation of CyP$_6$Q[6]@L-Leu@[ZnCl$_4$]$^{2-}$ crystals. CyP$_6$Q[6] (10 mg, 8.1 µmol), L-Leu (9.6 mg, 72.9 µmol), and ZnCl$_2$ · 2H$_2$O (5.6 mg, 32.4 $\mu$mol) were added to 3 M HCl (3 ml), and the solution was boiled for about 1 min. It was then filtered, and the filtrate was left to stand at room temperature. After several days, single crystals of CyP$_6$Q[6]@L-Leu@[ZnCl$_4$]$^{2-}$ suitable for X-ray diffraction analysis were obtained in 35% yield.

The introduction of CdCl$_2$ or ZnCl$_2$ into aqueous HCl solutions can produce [CdCl$_4$]$^{2-}$ or [ZnCl$_4$]$^{2-}$ etc. anions which is helpful to prepare various Q[$n$]-based single crystals owing to the outer surface interaction formed between these anions and the electrostatic potential positive outer surface of Q[$n$]s [2,32–33].

All of the obtained crystals were examined on a Bruker single-crystal diffractometer, and the crystallographic parameters are shown in the electronic supplementary material, table S1. The CCDC numbers of complexes **1–4** are 1 977 303–1 977 306, respectively.

## 2.3. Proton nuclear magnetic resonance spectroscopy

Gly, L-Leu, L-Lys and CyP$_6$Q[6] were each dissolved in D$_2$O. The amount of CyP$_6$Q[6] was kept fixed at 1 mM in all studies. The amino acid solution was added dropwise to the CyP$_6$Q[6] solution to excess, and proton nuclear magnetic resonance ($^1$H-NMR) spectra were recorded at appropriate intervals on a JEOL JNM-ECZ400s spectrometer at 25°C. D$_2$O was used as a field-frequency lock, and the observed chemical shifts are reported in parts per million (ppm) relative to D$_2$O as an internal standard ($\delta$ = 4.67 ppm).

## 2.4. Isothermal titration calorimetry

A $1.00 \times 10^{-4}$ mol l$^{-1}$ solution of CyP$_6$Q[6] in water (1.00 ml) was placed in the sample cell, and a $1.00 \times 10^{-3}$ mol l$^{-1}$ Gly solution was drawn into a 250 ml syringe. The temperature was set at 25°C, and the titration was conducted by adding 30 aliquots (6 µl per aliquot) of the Gly solution at intervals of 300 s.

A $1.00 \times 10^{-3}$ mol l$^{-1}$ solution of CyP$_6$Q[6] in water (1.00 ml) was placed in the sample cell, and a $1.00 \times 10^{-2}$ mol l$^{-1}$ L-Leu (L-Lys) solution was drawn into a 250 ml syringe. The temperature was set at 25°C, and the titration was conducted by adding 25 aliquots (10 µl per aliquot) of the L-Leu (L-Lys) solution at intervals of 300 s. The thermodynamic parameters of each system were determined on a Nano isothermal titration calorimetry (ITC) isothermal calorimeter. Considering that the top of the syringe is easy to mix in air bubbles, the data were analysed with ORIGIN 8.0 software using an independent model after deleting the first two unwanted data points.

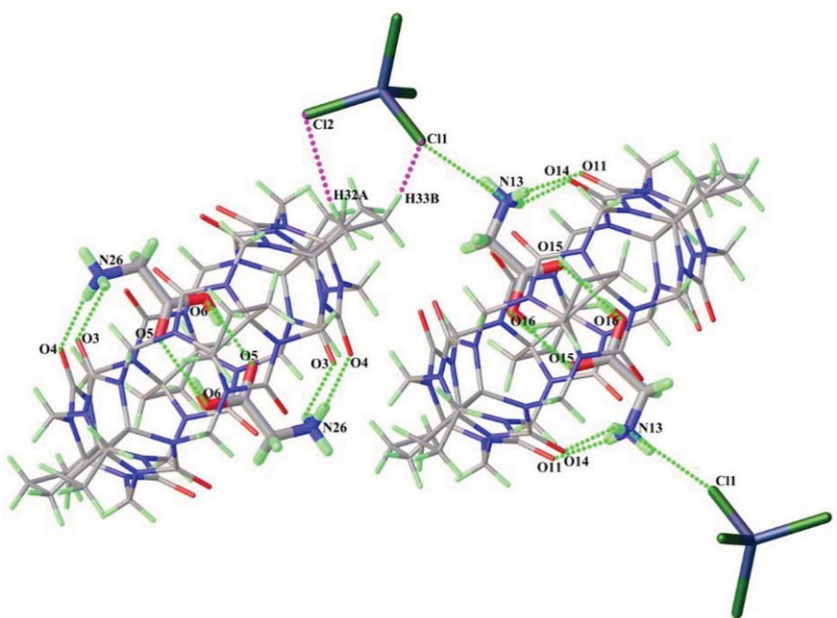

**Figure 1.** Host–guest interaction of complex **1**.

# 3. Results and discussion

## 3.1. Structural analysis of binding modes between CyP6Q[6] and amino acids

Figure 1 shows the crystal structure of Gly@CyP$_6$Q[6] (complex **1**). Analysis of the single-crystal structure shows that complex **1** belongs to the triclinic system with the centrosymmetric space group *P*-1. An oak ridge thermal-ellipsoid plot program (ORTEP) representation of the asymmetric unit is shown in the electronic supplementary material, figure S1. It contains two halves of CyP$_6$Q[6], two protonated glycine molecules and one free [CdCl$_4$]$^{2-}$ ion. In the single-crystal structure of complex **1**, each CyP$_6$Q[6] contains two glycine molecules. The carboxyl group of each glycine molecule is included in the cavity of the CyP$_6$Q[6], but its amino and methylene groups remain outside. The nitrogen atoms (N26 and N13) of the respective glycine molecules form two hydrogen bonds with two portal oxygen atoms (O5, O6 and O15, O16) of CyP$_6$Q[6], and the N–H $\cdots$ O distances are in the range 2.752–3.035 Å. It is interesting to note that a hydrogen bond is established between the nitrogen atom of a glycine molecule included by the cucurbituril and a chlorine atom (Cl1) of the counter ion [CdCl$_4$]$^{2-}$, with an N–H $\cdots$ Cl distance of 3.228 Å. This is not the case for the other amino acid molecule included by the cucurbituril. At the same time, there is also a dipolar interaction between this counter ion and a methylene proton on the outer wall of the other cucurbituril molecule. [CdCl$_4$]$^{2-}$ thus acts as a bridging unit to link the cucurbituril with the amino acid (shown by purple dotted lines in figure 1).

Figure 2 shows the crystal structure of L-Lys@CyP$_6$Q[6] (complex **2**). Analysis of the single-crystal structure shows that complex **2** belongs to the monoclinic system with centrosymmetric space group *P*21/*c*. An ORTEP representation of the asymmetric unit is shown in the electronic supplementary material, figure S2. It contains half of CyP$_6$Q[6], a protonated lysine molecule (occupancy ratio 0.5) and a free [CdCl$_4$]$^{2-}$ ion. In the single-crystal structure of complex **2**, each CyP$_6$Q[6] contains a lysine molecule. The carboxyl and amino groups of this lysine molecule, and the carbon atom (C2) to which they are bound, lie outside of the portal of CyP$_6$Q[6], while the rest of the molecule is within the cavity. The amino nitrogen atom (N1) and hydroxyl oxygen atom (O2) of the lysine molecule outside of the portal of CyP$_6$Q[6] form four hydrogen bonds (N1-H1C $\cdots$ O4, N1-H1A $\cdots$ O6, N1-H1C $\cdots$ O7, and O2-H2 $\cdots$ O7) with three oxygen atoms (O4, O6, and O7) of a portal of CyP$_6$Q[6], with lengths in the range 2.334–3.062 Å. The nitrogen atom (N2) of the terminal amino group of lysine inside the cavity forms a hydrogen bond with a portal carbonyl oxygen atom of the cucurbituril with a distance of 2.952 Å. Unlike in complex **1**, the counter ion does not interact with the amino acid in this complex, but only surrounds the cucurbituril through dipole interactions (electronic supplementary material, figure S5).

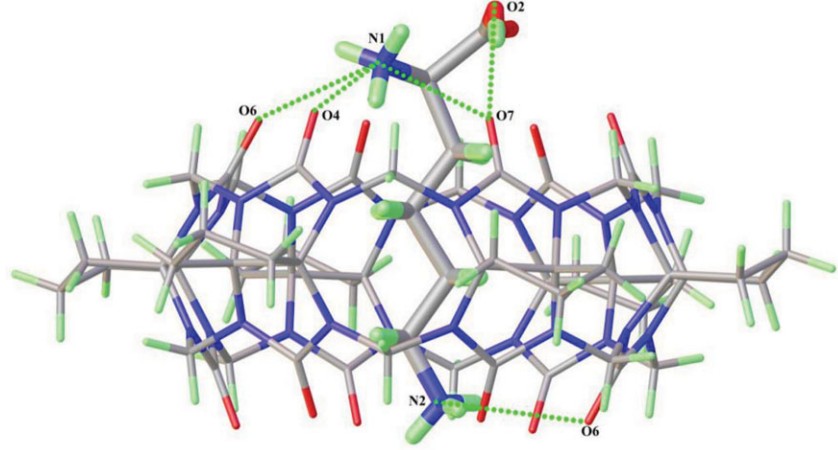

**Figure 2.** Host–guest interaction of complex **2**.

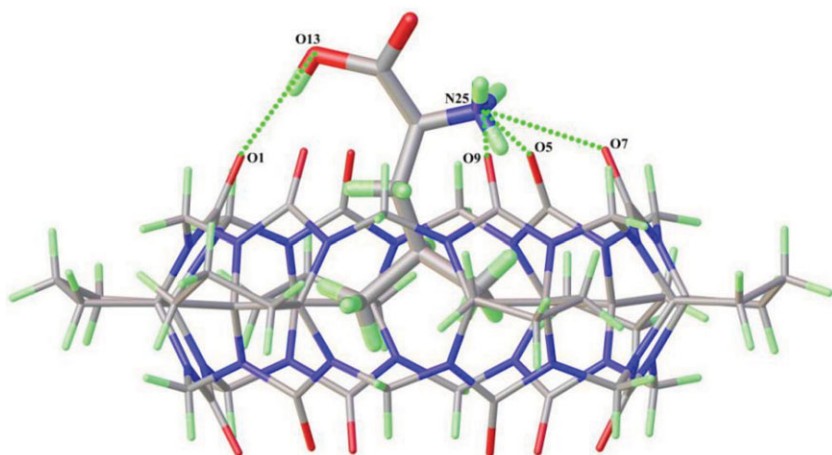

**Figure 3.** Binding mode of leucine and cucurbituril in complexes **3** and **4**.

Figure 3 shows the crystal structure of L-Leu@CyP$_6$Q[6] (complex **3**). Complexes **3** and **4** both comprise leucine and CyP$_6$Q[6]. Complex **3** belongs to the triclinic system with chiral space group $P1$, whereas complex **4** belongs to the monoclinic system with chiral space group $C2$. The main part of the asymmetric unit of these two complexes is composed of a CyP$_6$Q[6] host and a leucine molecule, and has the same binding mode. The difference is that the counter ion of complex **3** (electronic supplementary material, figure S3) is [ZnCl$_3 \cdot$ H$_2$O]$^-$, whereas that of complex **4** (electronic supplementary material, figure S4) is [ZnCl$_4$]$^{2-}$. Structural analysis of the main parts of these two complexes (taking complex **3** as an example; figure 3) shows that there is a hydrogen bond between the hydroxyl oxygen (O13) of the leucine molecule and portal carbonyl oxygen (O1) of the cucurbituril, with an O13–H13 ⋯ O1 distance of 2.629 Å. There are also hydrogen bonds between the amino nitrogen (N25) of the leucine molecule and three portal carbonyl oxygen atoms (O5, O7, O9) of the cucurbituril, with distances in the range 2.740–2.890 Å.

The counter ions of complex **3** are paired around the cucurbituril by ion-dipole interactions, and there is also an ion–dipole interaction between the two paired counter ions. Although the counter ions of complex **4** also surround the cucurbituril through ion-dipole interactions, there is no weak interaction between them. This difference results in very different stacking patterns of these two complexes. Figure 4 shows stack views of complexes **3** and **4** along the $c$-axis. From figure 4$a$,$b$, it can clearly be seen that all of the cucurbituril units in complex **3** have the same orientation, whereas those in complex **4** have two orientations with an included angle of 67.9°. Owing to the different orientations of the cucurbituril moieties, there is a significantly larger channel along the $c$-axis in complex **4** than in complex **3**.

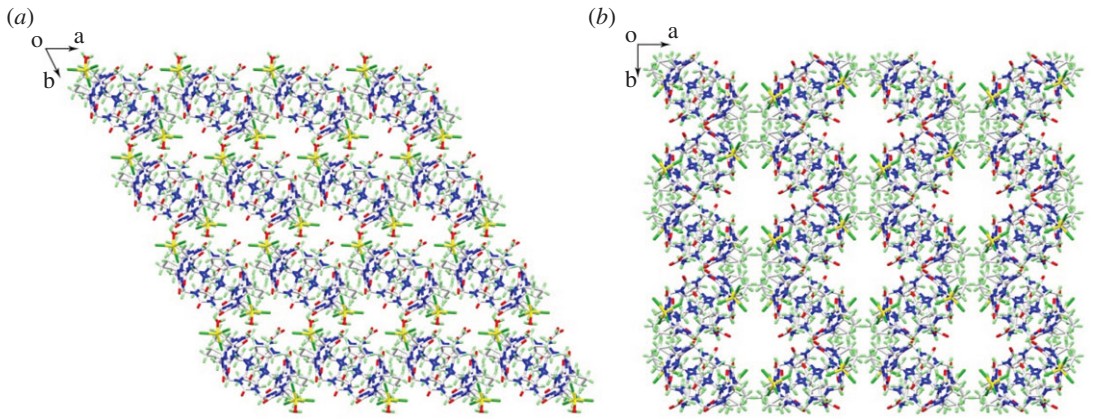

**Figure 4.** (a) Stack view of complex **3** along the c-axis; (b) stack view of complex **4** along the c-axis.

It is interesting to note that the carboxyl groups of the amino acids adopt a syn conformation in complex **1**, which appears in the crystal structure as the carboxyl group of glycine entering the cavity of CyP$_6$Q[6], and in complexes **2–4**, the carboxyl group of the amino acid exhibits anti-conformation, which appears in the crystal structure as the carboxyl group being outside the portal of the CyP$_6$Q[6]. This is mainly owing to the hydrophobic effect of the alkyl group. For complex **1**, there are only two hydrogens on the $\alpha$ carbon of glycine, and neither the group size nor the hydrophobic effect will affect the entry of the carboxyl group into the cucurbituril, showing a syn conformation. Compared with complex **1**, the $\alpha$ carbons of lysine and leucine of complexes **2–4** all have larger alkyl groups, and their hydrophobic effect is stronger than that of carboxyl groups and preferentially enter the cavity of CyP$_6$Q[6]. However, limited by the size of the cucurbituril cavity, the carboxyl groups are stuck on the outside of the cucurbituril, exhibiting anti conformation. It is worth noting that for the glycine using syn conformation, its size is relatively small compared with other amino acids, so that the cavity of CyP$_6$Q[6] is sufficient to accommodate two glycine molecules, and the carboxyl groups of the two amino acid molecules easily form hydrogen bonds in the hydrophobic cavity of CyP$_6$Q[6] to enhance their binding force, which can be proved in the binding constant part of the ITC experiment. However, because the carboxyl groups of leucine and lysine are exposed on the outside of the CyP$_6$Q[6], it is difficult to form hydrogen bonds between the carboxyl groups owing to the solvation effect on the outside of the CyP$_6$Q[6].

## 3.2. Proton nuclear magnetic resonance spectroscopy

CyP$_6$Q[6] shows good solubility in many solvents, most notably in water, in which it is one to two orders of magnitude more soluble than ordinary cucurbituril. Because the amino acids that make up the protein required for animal nutrition are mostly present in aqueous systems, the good water solubility of CyP$_6$Q[6] facilitates the study of its interactions with amino acids. In the present work, the binding behaviour of CyP$_6$Q[6] with the above three amino acids was investigated in D$_2$O. In $^1$H NMR, the cavity of CyP$_6$Q[6] has a shielding effect on proton signals, whereas outside of the portals, in the vicinity of the carbonyl oxygen atoms, the proton signals are subjected to a deshielding effect. According to this theory, the analysis of the $^1$H chemical shifts and splittings of the signals of protons of amino acids and the host provides insight into the binding mode between them. Figure 5 shows the changes in the $^1$H NMR spectra of the guest Gly as it is dropped into a solution of the host CyP$_6$Q[6]. The results show that the peak owing to the $\alpha$ protons of Gly shifts upfield, indicating that this unit enters the cavity of the cucurbituril. Considering the ion-dipole and hydrophobic effects, it may be speculated that the carboxyl group and methylene unit of Gly enter the cavity, while the amino group is fixed at the portal of CyP$_6$Q[6]. This binding mode is basically similar to the crystal structure. However, a difference is that the methylene unit lies outside of the portal in the solid phase, but inside the cavity in the liquid phase. At below two molar equivalents of Gly with respect to CyP$_6$Q[6], the $\alpha$ protons show only one signal, and the split triplet bridged methylene proton resonances (at 4.29 ppm) also show bound and free CyP$_6$Q[6] hosts. Beyond two molar equivalents, two signals due to this unit appear, corresponding to bound and free Gly, and the split triplet bridged methylene proton resonances turn to doublet again, suggesting that all CyP$_6$Q[6] molecules are

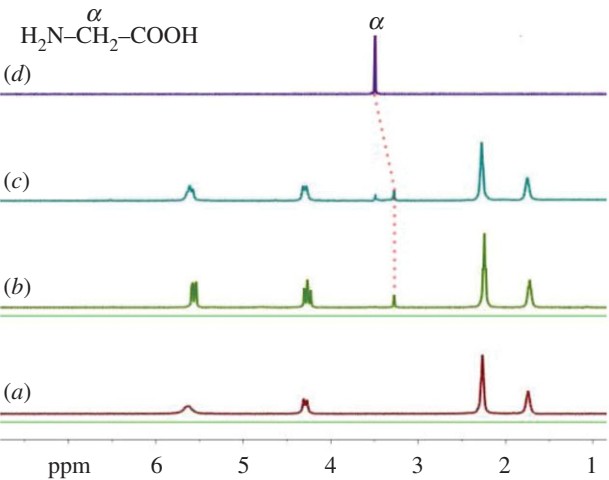

**Figure 5.** Titration $^1$H NMR spectra of Gly, in the presence of CyP6Q[6] (1 mM), with (a) 0.00, (b) 1.24, (c) 2.31 equiv. of Gly, and (d) neat Gly.

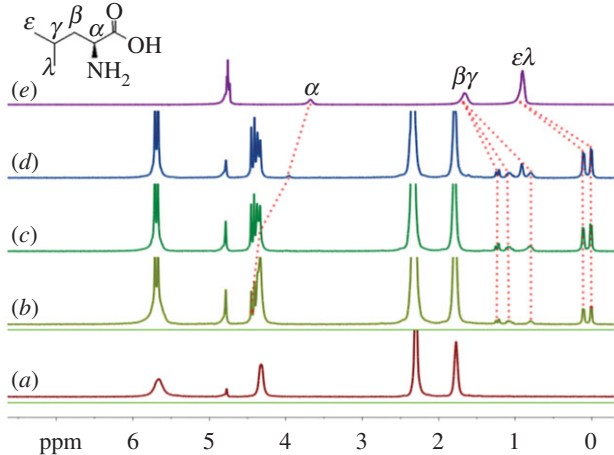

**Figure 6.** Titration $^1$H NMR spectra of L-Leu, in the presence of CyP6Q[6] (1 mM), with (a) 0.00, (b) 0.35, (c) 0.89, (d) 1.21 equiv. of L-Leu, and (e) neat L-Leu.

bound. This indicates that the exchange frequency is slower than the operating frequency of the $^1$H NMR spectrometer, based on the integral strength of the bound host and guest, which is closed to 1 : 2.0, suggesting that CyP$_6$Q[6] and Gly form a 1 : 2 inclusion complex, which is consistent with the solid phase (figure 5c).

The changes in the $^1$H NMR spectra of the guest L-Leu upon its incremental addition to CyP$_6$Q[6] are shown in figure 6. Two sets of proton resonances for L-Leu are observed (figure 6d), indicating that the exchange of bound and unbound L-Leu in the cavity of the CyP$_6$Q[6] host is slower than the operating frequency of the $^1$H NMR spectrometer, consistent with the observations for Gly. A difference is that the $\alpha$ proton signals of L-Leu shift downfield slightly, whereas the signals of the remaining alkyl chain protons shift upfield, suggesting that only the alkyl chain of a L-Leu guest enters the cavity of a CyP$_6$Q[6] molecule, and the carboxyl group remains outside of the portal. At the same time, it can be further seen in figure 6 that the $\varepsilon$ and $\lambda$ proton signals of the two methyl groups are split into two groups of signals from the original overlapping signal, indicating that these methyl groups are in different positions in the cavity of the CyP$_6$Q[6] host. A similar splitting is seen for the $\beta$ and $\gamma$ proton signals. After L-Leu interacts with the host, the environment of the two hydrogen protons of $\beta$ protons will have a certain difference, causing it to split into two sets of signals, so three peaks owing to the $\beta$ and $\gamma$ protons are seen. On the other hand, the split bridged methylene protons (at 4.32 ppm) of the bound CyP$_6$Q[6] host experience a slight down field shift. When a sub-stoichiometric amount of L-Leu is added to CyP$_6$Q[6], it displays only one set of signals. Once it is in excess, another set of signals owing to free L-Leu appears, based on the integral strength of the bound host and guest,

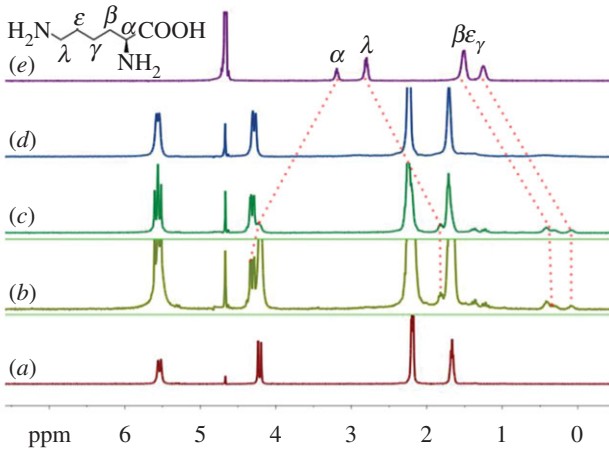

**Figure 7.** Titration $^1$H NMR spectra of L-Lys, in the presence of CyP6Q[6] (1 mM), with (*a*) 0.00, (*b*) 0.20, (*c*) 0.50, (*d*) 0.70 equiv. of L-Lys, and (*e*) neat L-Lys.

which is closed to 1 : 0.9, suggesting that $CyP_6Q[6]$ forms a 1 : 1 complex with L-Leu, as in the solid-phase crystal structure.

As shown in figure 7, titration $^1$H NMR spectroscopy was also used to investigate the binding behaviour between $CyP_6Q[6]$ and L-Lys. Unlike the above two cases, in which the bound guest, host and the free guest, host can be observed clearly in the titration process, when a sub-stoichiometric amount of L-Lys (0.2 and 0.5 equiv.) is added to $CyP_6Q[6]$, the bound guest, host and the free guest, host can be observed at the same time (figure 7*b*,*c*), in particular, when a sub-stoichiometric amount of L-Lys is up to 0.7 equiv. of $CyP_6Q[6]$, only proton resonances of the bound $CyP_6Q[6]$ can be observed clearly, and the proton resonances of the bound and unbound guests become broad and vague, suggesting a fast exchange of bound and unbound L-Lys in the cavity of the $CyP_6Q[6]$ host (figure 7*d*). Similarly to L-Leu, the $\alpha$ proton signals of L-Lys shift downfield, while the signals of the other alkyl chain protons shift upfield. This suggests that its alkyl chain enters the cavity of the $CyP_6Q[6]$, while the acid group remains outside the cavity, forming a structure similar to that of butanediamine@$CyP_6Q[6]$ [15], indicating that its alkyl chain enters the cavity of the host owing to the interaction of an extra amino group of L-Lys at the other portal of the host (referring to the structure in figure 2). However, we can estimate the interaction ratio of host : guest based on $^1$H NMR spectra for this particular case. The methine unit at which the carboxyl $\alpha$ proton is linked to one of the amino groups remains outside of the portal, forming a 1 : 1 complex of L-Lys@$CyP_6Q[6]$. The cavity of $CyP_6Q[6]$ is large enough to accommodate the L-Lys alkyl chain in its fully extended form, as corroborated by the crystal structure. It is worth noting that during the titration process, when the amount of L-Lys added reached 0.7 equivalents with respect to $CyP_6Q[6]$, its peaks suddenly broadened and even disappeared. This phenomenon may feasibly be attributed to the exchange frequency of L-Lys in and out of the cavity of the cucurbituril, exceeding the operating frequency of the $^1$H NMR spectrometer, such that the detected $^1$H NMR signals are averages of various intermediate states of the host–guest interaction. Averaging over a multitude of states will make the signals of the guest tend towards the baseline. Of course, it may also be the enrichment of lysine that causes the disappearance of the signal of the guest proton. The specific cause is still unclear, and further research is needed.

## 3.3. Isothermal titration calorimetry

ITC experiments (electronic supplementary material, figure S8) were performed to determine the thermodynamic parameters of the above three amino acids and $CyP_6Q[6]$ in water, providing insight into the thermal stability and driving force of the interactions. Table 1 shows that the enthalpies and entropies of the interactions of the three amino acids with $CyP_6Q[6]$ are both negative. From the contributions of these two thermodynamic parameters to the Gibbs free energy, it can be seen that the three systems are enthalpy-driven, and the driving force is determined by the ion-dipole interaction and the hydrophobic effect. The alkyl chain of the amino acid is more inclined to enter the cavity of the host owing to the hydrophobic effect, allowing water molecules originally in the cavity of the $CyP_6Q[6]$ to enter the aqueous phase, thereby reducing the entropy of the system. Moreover,

**Table 1.** Thermodynamic parameters of the interactions of amino acids with $CyP_6Q[6]$.

| experiment | $\Delta H$ (KJ/mol) | $T\Delta S$ (KJ/mol) | $K_a$ (M$^{-1}$) |
|---|---|---|---|
| Gly · $CyP_6Q[6]$ | −48.47 | −15.42 | $6.15 \times 10^5$ |
| L-Leu · $CyP_6Q[6]$ | −59.11 | −34.82 | $1.80 \times 10^4$ |
| L-Lys · $CyP_6Q[6]$ | −35.66 | −10.85 | $2.23 \times 10^4$ |

$2Gly@CyP_6Q[6]$ evidently has the largest binding constant among the three studied systems, which may be owing to the fact that there is some interaction between the two amino acids in addition to the interaction between glycine and cucurbituril. Its crystal structure (figure 1) shows that the carboxyl groups of both glycine molecules are also involved in hydrogen bonds, forming a more stable structure, so their binding constants are an order of magnitude higher than those for the other two amino acids. For lysine and leucine with the same number of carbon atoms, the binding constants are relatively close, but that of lysine is slightly higher owing to a dipolar interaction of the amino groups.

# 4. Conclusion

In summary, we have investigated the binding behaviour between $CyP_6Q[6]$ and three amino acids in both the solid and liquid phases. For lysine and leucine, both X-ray crystallography and $^1$H NMR spectroscopy indicate the formation of 1 : 1 host–guest complexes, with the respective alkyl chains within the cavity of $CyP_6Q[6]$. Glycine is bound slightly differently in the two phases; although a 1 : 2 inclusion complex is formed in each case, the methylene units have different locations. In the crystal structure, the methylene units lie outside of the portals, whereas solution $^1$H NMR shows that they lie within the cavity of the cucurbituril. ITC shows that the binding of all three amino acids is enthalpy-driven. Leucine shows two self-assembly modes, but this is not seen for the other two amino acids. The results of these experiments not only add to the understanding of the molecular recognition of amino acids, but are also of value for the design and synthesis of new bioactive cucurbiturils for the purpose of biological recognition and simulation.

Data accessibility. All data has been uploaded in the form of supporting information, and it has been noted in the manuscript.

Authors' contributions. S.Y.C. performed all laboratory work related to obtaining the compound and chemical assay, and wrote the highest percentage of the manuscript. W.W.Z. analysed and refined all crystal structures and wrote related content. X.N.Y. revised the manuscript. Y.M., assistant to S.Y.C., assisted in the synthesis and experimental analysis of raw materials. L.T.W. conducted the synthesis of the main substance. Z.T. guided the technical methods involved in the manuscript. P.H.M. was the leader of the project, holds the original idea, designed the statistical experiments and coordinated the experimental activities among all engaged laboratories. All authors read and approved the final manuscript.

Competing interests. We declare we have no competing interests

Funding. This work was supported by the National Natural Science Foundation of China (grant no. 21762011) and Guizhou Science and Technology Planning Project (Guizhou Science and Technology Cooperation Platform Talent [2017]5788), for collecting data, analysis and writing of this manuscript.

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
