## [Peer Review File · Royal Society Open Science]

Review History

RSOS-202120.R0 (Original submission)

Review form: Reviewer 1

Is the manuscript scientifically sound in its present form?

Yes

Are the interpretations and conclusions justified by the results?

Yes

Is the language acceptable?

Yes

Do you have any ethical concerns with this paper?

No

Have you any concerns about statistical analyses in this paper?

No

Recommendation?

Accept as is

Comments to the Author(s)

This manuscript is clearly presented with an interesting study of 3 amino acids showing significant binding in the cavity of, as the authors point out the 'immature' cyclopentanocucurbit[6]uril, with only a few example guests having been studied. The results are relevant and given that there is limited binding data for other comparable sized cucurbituril examples would suggest a greater significance of a binding advantage demonstrated for Cyp6Q[6]. The solid state evidence also provide a good comparison to the NMR results. This referee feels that the work should be published.

Review form: Reviewer 2**Is the manuscript scientifically sound in its present form?**

Yes

Are the interpretations and conclusions justified by the results?

No

Is the language acceptable?

Yes

Do you have any ethical concerns with this paper?

No

Have you any concerns about statistical analyses in this paper?

No

Recommendation?

Accept with minor revision (please list in comments)

Comments to the Author(s)

Ma and coworkers describe the crystal structures, NMR titrations, and isothermal titration calorimetry experiments for a cyclopentyl cucurbituril with three amino acids. The crystal structures are generally in agreement with the NMR data, and glycine behaves differently than lysine and leucine. The manuscript is interesting and I recommend publication after the following points are addressed by the authors.

1. In experimental section 2.4 (ITC), the volume of the aliquot is given in parentheses. For clarity, the authors should indicate if this is the volume per aliquot that is added or the total volume injected over the course of the experiment.
2. The last sentence of section 2.4 (ITC) states that the data was analyzed after the first two unwanted data points were deleted. The authors should clarify the reason why these data points were deleted.
3. Is there a reason cadmium salts were used in complexes 1 and 2, while zinc was used for complexes 3 and 4? This point should be clarified in the experimental section.
4. It is interesting to note that the carboxylic acid groups of the amino acids adopt a syn conformation in complex 1, an anti conformation in complex 2 and 4. In the anti conformation, the acid engages in hydrogen bonds with the cucurbituril, while in the syn conformation, it

engages in hydrogen bonds with another amino acid. A discussion about this point should be added to the main text.

5. The cif file of complex 3 includes hydrogen atoms bound to both oxygens of the carboxylic acid group in the amino acid molecule. However, the images in the main text and SI do not show both hydrogens. The cif file should be fixed to show the correct modeling of the carboxylic acid group.

6. In the discussion regarding the NMR titration of the cucurbituril with L-Lys (following Fig 6), the authors state that the alpha proton and all other alkyl chain protons shift upfield. However, in Fig 7, the alpha protons appear to shift downfield, analogous to what is observed in the titration with L-Leu. Thus, this suggests that the acid group remains outside the cavity, which is in agreement with the crystal structures.

Decision letter (RSOS-202120.R0)

Dear Dr Ma:

Title: The Binding Behaviors between Cyclopentanocucurbit[6]uril and Three Amino Acids
Manuscript ID: RSOS-202120

The editor assigned to your manuscript has now received comments from reviewers. We would like you to revise your paper in accordance with the referee and Subject Editor suggestions which can be found below (not including confidential reports to the Editor). Please note this decision does not guarantee eventual acceptance.

Please submit your revised paper before 05-Feb-2021. Please note that the revision deadline will expire at 00.00am on this date. If we do not hear from you within this time then it will be assumed that the paper has been withdrawn. In exceptional circumstances, extensions may be possible if agreed with the Editorial Office in advance. We do not allow multiple rounds of revision so we urge you to make every effort to fully address all of the comments at this stage. If deemed necessary by the Editors, your manuscript will be sent back to one or more of the original reviewers for assessment. If the original reviewers are not available we may invite new reviewers.

On behalf of the Subject Editor Professor Anthony Stace and the Associate Editor Dr Andrew Harned.

RSC Associate Editor: 1

Comments to the Author:

Both referees have expressed interest in this work. Through my own reading, I feel the manuscript is of high quality and will likely attract some interest. Reviewer 2 does raise a few valid concerns that should be addressed by the authors. I would also like to point out that one of the reviewers has raised a question with regard to how the NMR data of one of the complexes was interpreted. Such an interpretation may impact the overall conclusions of the manuscript. Please be sure to make appropriate changes as necessary.

RSC Associate Editor: 2

Comments to the Author:

(There are no comments.)

Reviewers' Comments to Author:

Reviewer: 1

Comments to the Author(s)

This manuscript is clearly presented with an interesting study of 3 amino acids showing significant binding in the cavity of, as the authors point out the 'immature' cyclopentanocucurbit[6]uril, with only a few example guests having been studied. The results are relevant and given that there is limited binding data for other comparable sized cucurbituril examples would suggest a greater significance of a binding advantage demonstrated for CyP6Q[6]. The solid state evidence also provide a good comparison to the NMR results. This referee feels that the work should be published.

Reviewer: 2

Comments to the Author(s)

Ma and coworkers describe the crystal structures, NMR titrations, and isothermal titration calorimetry experiments for a cyclopentyl cucurbituril with three amino acids. The crystal structures are generally in agreement with the NMR data, and glycine behaves differently than

lysine and leucine. The manuscript is interesting and I recommend publication after the following points are addressed by the authors.

1. In experimental section 2.4 (ITC), the volume of the aliquot is given in parentheses. For clarity, the authors should indicate if this is the volume per aliquot that is added or the total volume injected over the course of the experiment.
2. The last sentence of section 2.4 (ITC) states that the data was analyzed after the first two unwanted data points were deleted. The authors should clarify the reason why these data points were deleted.
3. Is there a reason cadmium salts were used in complexes 1 and 2, while zinc was used for complexes 3 and 4? This point should be clarified in the experimental section.
4. It is interesting to note that the carboxylic acid groups of the amino acids adopt a syn conformation in complex 1, an anti conformation in complex 2 and 4. In the anti conformation, the acid engages in hydrogen bonds with the cucurbituril, while in the syn conformation, it engages in hydrogen bonds with another amino acid. A discussion about this point should be added to the main text.
5. The cif file of complex 3 includes hydrogen atoms bound to both oxygens of the carboxylic acid group in the amino acid molecule. However, the images in the main text and SI do not show both hydrogens. The cif file should be fixed to show the correct modeling of the carboxylic acid group.
6. In the discussion regarding the NMR titration of the cucurbituril with L-Lys (following Fig 6), the authors state that the alpha proton and all other alkyl chain protons shift upfield. However, in Fig 7, the alpha protons appear to shift downfield, analogous to what is observed in the titration with L-Leu. Thus, this suggests that the acid group remains outside the cavity, which is in agreement with the crystal structures.

Author's Response to Decision Letter for (RSOS-202120.R0)

See Appendix A.

Decision letter (RSOS-202120.R1)

Dear Dr Ma:

Title: The Binding Behaviors between Cyclopentanocucurbit[6]uril and Three Amino Acids
Manuscript ID: RSOS-202120.R1

It is a pleasure to accept your manuscript in its current form for publication in Royal Society Open Science. The chemistry content of Royal Society Open Science is published in collaboration with the Royal Society of Chemistry.

On behalf of the Subject Editor Professor Anthony Stace and the Associate Editor Dr Andrew Harned.

RSC Associate Editor
Comments to the Author:
(There are no comments.)

Reviewer(s)' Comments to Author:

Appendix A

RE: Royal Society Open Science

Manuscript number: RSOS-202120

MS Type: Research Article

Title: "The Binding Behaviors between Cyclopentanocucurbit[6]uril and Three Amino Acids "

Correspondence Author: Professor Pei hua Ma

Email: phma@gzu.edu.cn.

Dear editor and reviewer:

Thank you very much for your valuable suggestions on this article. We have revised the manuscript and sent the revised manuscript for your consideration as a research paper to be published in Royal Society Open Science. Uploaded please find a manuscript entitled "The Binding Behaviors between Cyclopentanocucurbit[6]uril and Three Amino Acids". We have studied reviewer's comments carefully and have made revision which marked in red in the paper. We have tried our best to revise our manuscript according to the comments. Following are our responses (in BOLD type) to the Reviewers' comments.

Editor's comments:

RSC Associate Editor: 1

Comments to the Author:

Both referees have expressed interest in this work. Through my own reading, I feel the manuscript is of high quality and will likely attract some interest. Reviewer 2 does raise a few valid concerns that should be addressed by the authors. I would also like to point out that one of the reviewers has raised a question with regard to how the NMR data of one of the complexes was interpreted. Such an interpretation may impact the

overall conclusions of the manuscript. Please be sure to make appropriate changes as necessary.

Response: Thank you very much for your valuable suggestions. We have made appropriate changes in the manuscript based on the comments of reviewer 2.

RSC Associate Editor: 2

Comments to the Author:

(There are no comments.)

Response: Thank you for your reviewing our manuscript.

Reviewers' Comments to Author:

Reviewer: 1

Comments to the Author(s)

This manuscript is clearly presented with an interesting study of 3 amino acids showing significant binding in the cavity of, as the authors point out the ‘immature’ cyclopentanocucurbit[6]uril, with only a few example guests having being studied. The results are relevant and given that there is limited binding data for other comparable sized cucurbituril examples would suggest a greater significance of a binding advantage demonstrated for CyP₆Q[6]. The solid state evidence also provide a good comparison to the NMR results.

This referee feels that the work should be published.

Response: Thank you for agreeing to our paper. We hope to use a lot of examples in future study.

Reviewer: 2

Comments to the Author(s)

Ma and coworkers describe the crystal structures, NMR titrations, and isothermal titration calorimetry experiments for a cyclopentyl cucurbituril with three amino acids. The crystal structures are generally in agreement with the NMR data, and glycine behaves differently than lysine and leucine. The manuscript is interesting and I recommend publication after the following points are addressed by the authors.

1. In experimental section 2.4 (ITC), the volume of the aliquot is given in parentheses. For clarity, the authors should indicate if this is the volume per aliquot that is added or the total volume injected over the course of the experiment.

Response: This is a very good suggestion, We have made corresponding changes in the manuscript.

2. The last sentence of section 2.4 (ITC) states that the data was analyzed after the first two unwanted data points were deleted. The authors should clarify the reason why these data points were deleted.

Response: According to the reviewer's suggestion, we have made the following changes to the section:

Considering that the top of the syringe is easy to mix in air bubbles, the data were analyzed with ORIGIN 8.0 software using an independent model, after deleting the first two unwanted data points.

3. Is there a reason cadmium salts were used in complexes 1 and 2, while zinc was used for complexes 3 and 4? This point should be clarified in the experimental section.

Response: Thanks for the reviewer's suggestion. Indeed, in the process of cultivating the crystal structure, cadmium salts were used for complexes 1 and 2, while zinc salts for complexes 3 and 4. In fact, we all used cadmium salts at the beginning, but in the preparation of inclusion compound L-Leu@CyP6Q[6], after many attempts to use cadmium salts and failure to cultivate the corresponding crystals, we used zinc salts of the same group element and successfully cultivated crystals of complexes 3 and 4.

In this paper, both cadmium salts and zinc salts are used as inducers to promote the formation of self-assembly of inclusions to form crystal structures. Compared with this, we are more concerned about the binding behavior of CyP6Q[6] with amino acids. Of course, how various salt solutions affect their self-assembly behavior and whether they affect the interaction mode of the host-guest of the cucurbituril is a focus of future research in our laboratory.

4. It is interesting to note that the carboxylic acid groups of the amino acids adopt a syn conformation in complex 1, an anti conformation in complex 2 and 4. In the anti conformation, the acid engages in hydrogen bonds with the cucurbituril, while in the syn conformation, it engages in hydrogen bonds with another amino acid. A discussion about this point should be added to the main text.

Response: This is a very good suggestion. We have added related content in manuscript, as follows:

It is interesting to note that the carboxyl groups of the amino acids adopt a syn conformation in complex 1, which appears in the crystal structure as the carboxyl group of glycine entering the cavity of CyP₆Q[6]. And in complex 2~4, the carboxyl group of the amino acid exhibits anti conformation, which appears in the crystal structure as the carboxyl group being outside the portal of the CyP₆Q[6]. This is mainly due to the hydrophobic effect of the alkyl group. For complex 1, there are only two hydrogens on the α carbon of glycine, and neither

the group size nor the hydrophobic effect will affect the entry of the carboxyl group into the cucurbituril, showing a syn conformation. Compared with complex 1, the α carbons of lysine and leucine of complex 2~4 all have larger alkyl groups, and their hydrophobic effect is stronger than that of carboxyl groups and preferentially enter the cavity of CyP₆Q[6]. However, limited by the size of the cucurbituril cavity, the carboxyl groups are stuck on the outside of the cucurbituril, exhibiting anti conformation. It is worth noting that for the glycine using syn conformation, its size is relatively small compared with other amino acids, so that the cavity of CyP₆Q[6] is sufficient to accommodate 2 glycine molecules, and the carboxyl groups of the two amino acid molecules easily form hydrogen bonds in the hydrophobic cavity of CyP₆Q[6] to enhance their binding force, which can be proved in the binding constant part of the ITC experiment. However, because the carboxyl groups of leucine and lysine are exposed on the outside of the CyP₆Q[6], it is difficult to form hydrogen bonds between the carboxyl groups due to the solvation effect on the outside of the CyP₆Q[6].

5. The cif file of complex 3 includes hydrogen atoms bound to both oxygens of the carboxylic acid group in the amino acid molecule. However, the images in the main text and SI do not show both hydrogens. The cif file should be fixed to show the correct modeling of the carboxylic acid group.

Response: We would like to apologize for our carelessness. We have modified and replaced the cif file as required.

6. In the discussion regarding the NMR titration of the cucurbituril with L-Lys (following Fig 6), the authors state that the alpha proton and all other alkyl chain protons shift upfield. However, in Fig 7, the alpha protons appear to shift downfield, analogous to what is observed in the titration with L-Leu. Thus, this suggests that the acid group remains outside the cavity, which is in agreement with the crystal structures.

Response: Thank you very much for your pertinent suggestions on this article. We have tried our best to make the corresponding changes based on your valuable suggestions in the text. The details are as follows:

Similarly to L-Leu, the α proton signals of L-Lys shift downfield, while the signals of the other alkyl chain protons shift upfield. This suggests that its alkyl chain enters the cavity of the CyP₆Q[6], while the acid group remains outside the cavity, forming a structure similar to that of butanediamine@CyP₆Q[6],^[15] indicating that its alkyl chain enters the cavity of the host due to the interaction of an extra amino group of L-Lys at other portal of the host.

We would like to express our great appreciation to you and reviewers for comments on our paper. Looking forward to hearing from you.

Thank you and best regards.

Yours sincerely,

Pei hua Ma

Key Laboratory of Macrocyclic and Supramolecular Chemistry of Guizhou Province,
Guizhou University, Guiyang 550025, People's Republic of China.

E-mail: phma@gzu.edu.cn

Tel: 86-13368690478